# Screening Mammography and Breast Cancer: Variation in Risk with Rare Deleterious or Predicted Deleterious Variants in DNA Repair Genes

**DOI:** 10.3390/cancers17071062

**Published:** 2025-03-21

**Authors:** Maximiliano Ribeiro-Guerra, Marie-Gabrielle Dondon, Séverine Eon-Marchais, Dorothée Le Gal, Juana Beauvallet, Noura Mebirouk, Muriel Belotti, Eve Cavaciuti, Claude Adenis-Lavignasse, Séverine Audebert-Bellanger, Pascaline Berthet, Valérie Bonadona, Bruno Buecher, Olivier Caron, Mathias Cavaille, Jean Chiesa, Chrystelle Colas, Isabelle Coupier, Capucine Delnatte, Hélène Dreyfus, Anne Fajac, Sandra Fert-Ferrer, Jean-Pierre Fricker, Marion Gauthier-Villars, Paul Gesta, Sophie Giraud, Laurence Gladieff, Christine Lasset, Sophie Lejeune-Dumoulin, Jean-Marc Limacher, Michel Longy, Alain Lortholary, Elisabeth Luporsi, Christine M. Maugard, Isabelle Mortemousque, Sophie Nambot, Catherine Noguès, Pascal Pujol, Laurence Venat-Bouvet, Florent Soubrier, Julie Tinat, Anne Tardivon, Fabienne Lesueur, Dominique Stoppa-Lyonnet, Nadine Andrieu

**Affiliations:** 1Inserm, U1331, 75248 Paris cedex 05, France; guerramr@hotmail.com (M.R.-G.); marie-gabrielle.dondon@curie.fr (M.-G.D.); severine.eonmarchais@curie.fr (S.E.-M.); dorothee.legal@curie.fr (D.L.G.); juana.beauvallet@curie.fr (J.B.); noura.mebirouk@gmail.com (N.M.); eve.cavaciuti@curie.fr (E.C.); fabienne.lesueur@curie.fr (F.L.); 2Institut Curie, 75248 Paris cedex 05, France; 3Mines ParisTech, 75272 Paris cedex 06, France; 4PSL Research University, 75006 Paris, France; 5Department of Public Health, Faculty of Medicine, Federal University of Juiz de Fora (UFJF), Juiz de Fora 36036-900, MG, Brazil; 6Institut Curie, Service de Génétique, 75248 Paris cedex 05, France; muriel.belotti@curie.fr (M.B.); bruno.buecher@curie.fr (B.B.); chrystelle.colas@curie.fr (C.C.); marion.gauthier-villars@curie.fr (M.G.-V.); dominique.stoppa-lyonnet@curie.fr (D.S.-L.); 7Polyclinique de la Louvière (Groupe Ramsay), 59800 Lille, France; c.adenis@ramsaygds.fr; 8Département de Génétique Médicale et Biologie de la Reproduction, Hôpital Morvan, CHU Brest, 29200 Brest, France; severine.audebert@chu-brest.fr; 9Unité de Pathologie Gynécologique, Centre François Baclesse, 14000 Caen, France; p.berthet@baclesse.fr; 10CNRS, Laboratoire de Biométrie et Biologie Evolutive UMR 5558, Université Lyon 1, 69622 Villeurbanne, France; valerie.bonadona@lyon.unicancer.fr (V.B.); christine.lasset@lyon.unicancer.fr (C.L.); 11Centre Léon Bérard, Unité de Prévention et Epidémiologie Génétique, 69008 Lyon, France; 12Gustave Roussy, Département de Médecine Oncologique, 94800 Villejuif, France; olivier.caron@gustaveroussy.fr; 13Centre Jean Perrin, Département d’Oncogénétique, Inserm, UMR 1240, Université Clermont Auvergne, 63001 Clermont Ferrand, France; mathias.cavaille@clermont.unicancer.fr; 14Service d’Oncologie Médicale, CHRU Hôpital Caremeau, 30900 Nîmes, France; jean.chiesa@chu-nimes.fr; 15Institut Curie, Hôpital René Huguenin, 92210 Saint-Cloud, France; 16Service de Génétique Médicale et Oncogénétique, Hôpital Arnaud de Villeneuve, CHU Montpellier, 34090 Montpellier, France; coupier@chu-montpellier.fr (I.C.); p-pujol@chu-montpellier.fr (P.P.); 17Inserm, U896, CRCM Val d’Aurelle, 34090 Montpellier, France; 18Unité d’Oncogénétique, Institut de Cancérologie de l’Ouest, 44800 Saint-Herblain, France; capucine.delnatte@ico.unicancer.fr; 19Clinique Sainte Catherine, 84000 Avignon, France; h.dreyfus@isc84.org; 20Département de Génétique, Hôpital Couple-Enfant, CHU de Grenoble, 38700 Grenoble, France; 21Service d’Oncogénétique, Hôpital Tenon, 75020 Paris, France; anne.fajac@tnn.aphp.fr; 22Service de Génétique, Centre Hospitalier Métropole Savoie, 73000 Chambéry, France; sandra.fertferrer@ch-chambery.fr; 23Centre Paul Strauss, Unité d’Oncologie, 67200 Strasbourg, France; jeanpierrefricker@sfr.fr; 24Service d’Oncogénétique Régional Poitou-Charentes, CH Georges Renon, 79000 Niort, France; paul.gesta@ch-niort.fr; 25Centre Hospitalier de la Côte Basque, Institut Bergonié, 33000 Bordeaux, France; sophie.giraud@chu-lyon.fr; 26Service d’Oncologie Médicale, Institut Claudius Regaud, IUCT-Oncopole, 31100 Toulouse, France; gladieff.laurence@iuct-oncopole.fr; 27Clinique de Génétique Médicale Guy Fontaine, CHU Lille, 59000 Lille, France; sophie.lejeune@chru-lille.fr; 28Service d’Onco-Hématologie, Hôpital Pasteur, 68024 Colmar, France; jean-marc.limacher@ch-colmar.fr; 29Cancer Genetics Unit, Inserm U1312, Institut Bergonié, University of Bordeaux, 33000 Bordeaux, France; m.longy@bordeaux.unicancer.fr; 30Service d’Oncologie Médicale, Centre Catherine de Sienne, 44200 Nantes, France; alain.lortholary@groupeconfluent.fr; 31Service de Génétique, UF4128, CHR Metz-Thionville, Hôpital de Mercy, 57530 Ars-Laquenexy, France; e.luporsi@chr-metz-thionville.fr; 32UF1422, Génétique Oncologique Moléculaire, Département d’Oncobiologie, LBBM, Hôpitaux Universitaires de Strasbourg, 67200 Strasbourg, France; christine.maugard@chru-strasbourg.fr; 33UF6948, Génétique Oncologique Clinique, Evaluation Familiale et Suivi, Hôpitaux Universitaires de Strasbourg, 67200 Strasbourg, France; 34Service de Génétique, Hôpital Bretonneau, 75018 Paris, France; i.mortemousque@chu-tours.fr; 35Institut GIMI, Oncogénétique, CHU de Dijon, Hôpital d’Enfants, 21000 Dijon, France; sophie.nambot@chu-dijon.fr; 36Centre de Lutte Contre le Cancer Georges François Leclerc, 21000 Dijon, France; 37Département d’Anticipation et de Suivi des Cancers, Oncogénétique Clinique, Institut Paoli-Calmettes, 13009 Marseille, France; noguesc@ipc.unicancer.fr; 38Inserm, IRD, SESSTIM, Aix Marseille University, 13009 Marseille, France; 39Service d’Oncologie Médicale, Hôpital Universitaire Dupuytren, 87000 Limoges, France; laurence.venat@chu-limoges.fr; 40UMR_S 1166, Faculté de Médecine, SU Site Pitié-Salpêtrière, 75013 Paris, France; dr.florent.soubrier@gmail.com; 41Service de Génétique Médicale, CHU De Bordeaux, Groupe Hospitalier Pellegrin, 33000 Bordeaux, France; julie.tinat@chu-bordeaux.fr; 42Service de Radiologie, Institut Curie, 75005 Paris, France; anne.tardivon@curie.fr; 43Inserm, U830, 75005 Paris, France; 44Université Paris-Cité, 75006 Paris, France

**Keywords:** breast cancer, mammography screening, high-risk population, DNA repair genes

## Abstract

The association of screening mammography with breast cancer (BC) was investigated in cases with a hereditary predisposition unexplained by *BRCA1* or *BRCA2* and unrelated controls. Participants reported their lifetime mammography exposures in a questionnaire. Additionally, germline rare deleterious or predicted deleterious variants (D-PDVs) were investigated in 113 DNA repair genes. No association was found between having been exposed to mammograms (never vs. ever) and BC. However, when considering the number of mammograms, an increase in BC risk of 4% (95% CI: 1–6%) per additional exposure was found. When women were grouped according to their D-PDV carrier status and the estimated associated BC risk, mammograms doubled the BC risk of women carrying a D-PDV in a gene associated with BC with an odds ratio (OR) < 0.9, as compared to those carrying a D-PDV in a gene with an OR > 1.1. Even though mammographic screening reduces the risk of mortality from BC, the identification of populations more or less susceptible to ionizing radiation may be clinically relevant.

## 1. Introduction

Women with a genetic predisposition to breast cancer (BC) may be at increased risk of cancer after exposure to ionizing radiation. Screening mammography is associated with exposure of the breast to a small dose of radiation (i.e., the regulatory limit set at 3.0 mGy for each view of an average-sized breast and two views of each breast), which has not been found to increase BC risk in the general population [1]. However, women who are at increased genetic risk might be more sensitive to the DNA-damaging effect of ionizing radiation, in particular women carrying a deleterious variant in a gene involved in the repair of DNA damages like *BRCA1* and *BRCA2* [2]. Moreover, women with a family history of BC are offered screening at earlier ages and at more frequent intervals than women from the general population. Assuming there is no lower threshold dose in which radiation exposure does not cause damage, repeated exposure to ionizing radiation from screening mammographic procedures may increase the risk of BC, and the possible benefit of mammography screening could be reduced due to the risk of radiation-induced tumors.

While exposures to diagnostic ionizing radiation to the chest excluding mammograms have been studied in the population with a genetic predisposition to BC [2,3,4,5,6,7,8], few studies have investigated the effect of mammograms in such populations, and these studies were essentially conducted among *BRCA1/2* pathogenic variant carriers [2,4,9,10,11,12,13]. Therefore, we evaluated the effect of radiation exposure from mammography screening on BC risk in women with a genetic predisposition to BC and tested negative for a *BRCA1/2* pathogenic variant. Furthermore, we evaluated whether carrying a rare deleterious or predicted deleterious variant (D-PDV) (i.e., loss-of-function or missense variant with a predicted deleterious effect, phred CADD score ≥ 20 [14]) in 113 DNA repair genes (other than *BRCA1/2*) previously selected [15,16] modifies the association between mammography screening exposures and BC risk.

## 2. Methods

### 2.1. Study Population

The study population involved cases and unrelated controls enrolled in GENESIS between February 2007 and December 2013, a national study initially set up to investigate genetic factors involved in the susceptibility to BC other than *BRCA1/2* [17]. Index cases were identified through the French family cancer clinics of the Groupe Génétique et Cancer (Unicancer) (i.e., 42 centers) and were eligible when diagnosed with infiltrating mammary or ductal adenocarcinoma, negative for *BRCA1* and *BRCA2* mutations, and had a sister with BC (Figure 1). The mutation screening strategy was similar for all the clinics (cf. Appendix A). The unrelated controls were selected from among the unaffected friends and/or colleagues of the cases. The year of birth of controls was matched to that of the case (±3 years). All women completed a questionnaire on environmental, lifestyle, and reproductive factors and family history of cancer. Only women reporting European ancestry (i.e., over 95% of the study population) were considered for this study.

### 2.2. Screening Mammography

Participants reported their history of screening mammography in a detailed questionnaire at the time of their recruitment. To exclude mammograms that could have been performed because of BC diagnosis, we considered exposures that occurred up to one year prior to BC diagnosis for cases and one year prior to the date of questionnaire completion for controls. Variables considered in the analyses were ever vs. never exposed, lifetime number of screening mammograms, age at first mammography, and timing since the first mammography, i.e., the delay in years between the age at first mammography and the age at BC diagnosis for cases and the age at interview for the controls (i.e., age at censuring).

We excluded 52 women who underwent radiotherapy for a benign disease more than one year prior to age at censoring (2.19% cases; 1.23% controls). Among cases, we also excluded 10 women (0.63%) who underwent radiotherapy for a cancer other than BC before their BC diagnosis.

### 2.3. Variants in DNA Repair Genes

Contribution of germline rare D-PDVs (with minor allele frequency <0.5% in controls) in 113 DNA repair genes in familial BC was previously assessed by performing targeted sequencing of their entire coding sequence in 1207 cases and 1199 controls from the GENESIS study. Detailed information on the selection of genes, sequencing procedure, and variants filtering and annotation is described in Girard et al. [16]. Loss-of-function and missense variants with a phred CADD score ≥ 20 as a predictor of deleterious effect [14] and with minor allele frequency <0.5% in controls were selected and defined as D-PDV. Sequencing data were available for 82.5% of the women investigated in the present study, and no difference in the distribution of the characteristics such as education level, smoking, or ionizing exposures was observed between the subsets of cases and controls with and without sequencing data (see Supplemental Table 1 in Ribeiro-Guerra et al. [15]).

Because study participants carried very few rare D-PDVs per gene (pool of variants for each gene ranged between 0% and 4.1% in controls), we grouped the genes according to the value of their association with BC, i.e., the odds-ratio (OR) point estimate obtained in the study by Girard et al. [16] (whatever the degree of significance) and classified them as follows: Group “Reduced” including genes with OR < 0.9; Group “Independent”, including genes with 0.9 ≤ OR ≤ 1.1; and Group “Increased”, including genes with OR > 1.1. An individual could be assigned to more than one group if carrying several variants in genes belonging to different groups [15]. Results of the association tests per gene were first published [16]. Gene group assignation is defined in Ribeiro-Guerra et al. [15] and reported in Appendix A to ease the reading. Additionally, we hypothesized that variants might have additive effects and built a Variant Risk Score (VRS = β_1_x_1_ + β_2_x_2_ +…+ β_113_x_113_ with β the per DNA repair gene minor allele mean effect, i.e., log OR for BC, and x equals 1 or 0 for being carrier or non-carrier, respectively) and performed analysis using the above bounds for ORs.

### 2.4. Statistical Analyses

To assess the association between mammography exposure and BC risk, we used logistic regression models adjusted for variables described in Appendix A.

We assessed associations between mammography exposure and BC by DNA repair gene group and used likelihood ratio tests to test for heterogeneity and multiplicative interaction. Additionally, we adjusted for other gene groups when the analysis was stratified by gene group.

We assessed heterogeneity between ER tumor status using a multinomial logistic regression model and tested the equality of coefficients between equations by the difference between the log-likelihoods.

All *p*-values using Z-tests were two-sided, and a 5% level of significance was used. All analyses were performed using Stata software version 14 [18].

## 3. Results

Characteristics of the study population are described in Table 1. “Most of the cases were prevalent with a mean delay between diagnosis and interview of 8.3 years (SD: ±7.1). The mean age at BC diagnosis was 50.2 years (SD: ±9.3) and the mean age at interview for the controls was 55.8 years (SD: ±9.9)” as described in Ribeiro-Guerra et al. [15].

Compared to controls, index cases were more likely to have a basic education level, lower body mass index, and as expected, a stronger family history of BC at censure. Regarding birth cohort, cases were more likely to be born before 1945 than controls. Among the subset of participants who had been sequenced for the 113 DNA repair genes (74.1%), 20% had no D-PDV, 31% carried 1 D-PDV, 26%, 2 D-PDVs, and 23% carried 3 or more D-PDVs. Cases were more likely to carry a variant in a gene from the “Increased” risk group than controls (57.7% and 42.6%, respectively) (Table 1). Mammography exposure characteristics are described in Table 2. More controls had mammograms than cases (91.9% and 81.9%, respectively). The mean number of exposures was similar for cases and controls, and the first exposure occurred earlier for cases than for controls (38.9 years and 43.1 years, respectively).

As a whole, we did not find an association between BC and having been exposed to mammograms when measured as ever vs. never (OR = 0.99; 95% CI: 0.69–1.43) (Table 3) or when categorized according to the number of mammograms. However, when the number of mammograms was considered as a continuous variable, under the assumption of linearity, we found a significant trend in BC risk (OR = 1.04; 95% CI: 1.01–1.06) with an increased risk of 4% by additional exposure. We did not find a significant association between BC and age at the first mammogram exposure and time since first exposure although we found an OR point estimate increasing as the age at first exposure decreases from age 40 to age 30 (OR = 1.52; 95% CI: 0.91–2.53). Among exposed women, under the assumption of linearity, we found a significant decrease in BC risk of 2% per additional year for age at first exposure (95% CI: 1–4%).

When analyses were performed according to the estrogen (ER) status of the tumor (Table 4), there were no differences in ORs between ER− and ER+ tumors for almost all mammogram exposure variables except for the number of mammograms, with a significant trend observed for women with ER+ tumors (8% per additional exposure, 95% CI: 4–11%).

As expected, having at least one D-PDV in a gene from Group “Reduced” is associated with a significant decrease in BC risk of 0.62 (95% CI: 0.50–0.77) compared to not having D-PDV in a gene from Group “Reduced”; having at least one D-PDV in a gene from Group “Independent” is associated with a non-significant BC risk of 1.02 (95% CI: 0.84–1.25), and having at least one D-PDV in a gene from Group “Increased” is associated with a significant increased BC risk of 1.98 (95% CI: 1.62–2.42) compared to not having D-PDV in a gene from Group “Increased” (Table 5). When stratifying women according to the group of altered genes, we found a difference in the association between having been exposed to mammograms and BC but not in the expected direction. The effect of mammograms’ exposure on BC risk was significantly higher for women carrying at least one variant in Group “Reduced” than for those carrying at least one variant in the other groups and especially in Group “Increased” (P_int_ = 0.02) (Table 6) (ever vs. never: “Reduced”_(OR<0.9)_: OR = 2.17; “Independent”_(0.9≤OR≤1.1)_: OR = 1.25; “Increased”_(OR>1.1)_: OR = 0.92).

In the three groups, there was no significant association with the categorized number of mammograms although a significant trend was found, with an increased risk of 6% (95% CI: 2–10%) by additional exposure for both Group “Independent” and Group “Increased”. Additionally, we found significant multiplicative interactions for Group “Increased” (P_int_ = 0.020 and 0.019 for 1 to 4 and 5 to 9 exposures, respectively). Having been first exposed before age 30 increased the risk of BC significantly only in Group “Reduced” with an OR of 3.16 (95% CI: 1.01–9.89). For the three other age classes at first exposure, we found significant interactions for Group “Increased” (P_int_ = 0.019, 0.021, and 0.057, respectively). As for the whole study population, we did not find an association between BC and time since first exposure in any gene group. Additionally, as ~80% of participants carried more than one D-PDV (Table 1) and could be categorized into more than one DNA repair gene group, we hypothesized that variants might have additive effects and built a DNA repair VRS (Figure 2) and estimated the risk of BC per quintile of VRS. As expected, BC risk increased as VRS increased (Table 7). When stratifying women according to VRS quintiles, results for association between mammograms and BC confirmed the trend observed in the previous analyses, i.e., a decrease in BC risk associated with mammograms as DNA repair VRS increased (<ln(0.8): OR = 2.56; ≥ln(0.8) & <ln(0.95): OR = 3.88; ≥ln(0.95) & <ln(1.1): OR = 2.29; ≥ln(1.1) & <ln(1.5): OR = 1.24; ≥ln(1.5): OR = 0.30; P_int_ = 0.006) (Table 7). The same trend is observed as VRS increased when first exposed after age 40 or before age 40, but with point estimates higher for the latest group of women (Table 7).

Because we used a priori bounds for ORs for defining the DNA repair gene groups, we performed several sensitivity analyses using different bounds, and we found similar trends in the difference in the BC risk between groups (Appendix A).

We also performed a sensitivity analysis that included in the “Increased” Group only the genes that were found significantly (or borderline) associated with BC in our population [16] (i.e., *ATM*, *CHEK2*, *PALB2*, *FANCM*, *MAST1*) or confirmed as pathogenic or predicted to be pathogenic for BC by international consortia [19,20] (i.e., *PALB2*, *TP53*, *CHEK2*, *PTEN*, *ATM*, *BARD1*, *MSH6*, *RAD51C*, *RAD51D*). Only a few participants had never been exposed to mammography among those D-PDV carriers (28 and 24, respectively), and similar point estimates are however observed (ever vs. never: OR = 0.32; 95% CI: 0.07–1.54 and OR = 0.22; 95% CI: 0.04–1.23, respectively).

Finally, the magnitude and direction of the effect estimates based on analyses using an extra class for missing data or a multiple imputation strategy were similar (Table 3 and Table 8, Appendix A).

## 4. Discussion

Overall, we found that mammography exposure, measured as ever vs. never, is not associated with the risk of developing a BC in women with a hereditary predisposition to BC unexplained by a *BRCA1* or *BRCA2* pathogenic variant. However, this risk increases with the number of mammograms, with an increase of 4% for each additional exposure. An increased risk is also suggested when first exposure occurs before age 40. To our knowledge, only three studies investigated diagnostic radiation exposure to the chest among women with a hereditary predisposition to BC and almost all without *BRCA1/2* pathogenic variant [5,15,21], and only one considered mammograms by assessing the effect of age at first exposure and did not find evidence for an increased BC risk for women exposed before 37 years of age compared to women first exposed between the ages of 40 and 46 years [21]. Only four studies assessed the effect of a history of mammography among *BRCA1/2* pathogenic variant carriers [2,4,9,10,11,12,13] and only one found an increased point estimate with a hazard ratio of 1.43 for women first exposed before age 30 [2], which is in line with our findings.

Surprisingly, our findings suggested that radiation delivered by mammograms double the risk of BC in women carrying a D-PDV in a DNA repair gene negatively associated with the risk of BC, while such an exposure appears not to be associated with BC in women carrying a D-PDV in a gene associated with an increased risk of BC. This finding is counterintuitive and not in line with our previous study [15], where we found a difference in the association between having been exposed to chest X-ray and BC with a significantly higher risk for women carrying at least one D-PDV in DNA repair genes associated with an increased risk of BC.

A possible weakness of our study is the use of nonrandomly selected friends or colleagues as controls. However, this approach offered the advantage of making it easier to identify suitable controls compared to random selection from the general population, and a higher comparability for unmeasured factors with, however, a likely indication bias for mammograms. Indeed, a strong family history of BC could be an indication for mammographic screening, especially at a young age. We investigated this potential bias away from the null by looking at the distribution of the family history by category of age at first exposure and did not find difference (chi2 = 23.3, df = 23, *p* = 0.438) (Appendix A).

However, to take into account potential confounders and indication biases, we adjusted the analyses for numerous variables like age at censure, birth cohort, being eligible for national screening, and the number of BC in the family at censure.

Because the mammography technique has evolved over time, as has the level of dose received for each image, we are aware that an estimation of the doses received at each examination according to when and where the examination was carried out would have generated more precise exposure data. However, retrospective studies are not able to trace and standardize mammography procedures through medical record reviews because of the difficulties in accessing medical records. For this reason, information on lifetime mammograms’ exposures was self-reported with accompanying potential recall biases and exposure misclassification (see also Appendix A). Therefore, even if some studies have shown that self-reported mammography data were widely reliable for epidemiologic research [22,23], we cannot totally rule out such a bias. Thus, to avoid the potential effect of exposure misclassification, indication, and recall biases, we performed case-only analyses and assessed the interaction effects between the groups of D-PDVs and mammograms’ exposure (under the assumption of independence). We found an OR_interaction_ = 1.62 (*p* = 0.10) between mammograms’ exposure and DNA repair genes potentially decreasing BC risk, OR_interaction_ = 0.84 (*p* = 0.48) between mammograms’ exposure and those with no effect, and OR_interaction_ = 0.68 (*p* = 0.14) between mammograms exposure and those with increasing BC risk). Even though not statistically significant, these results are in line with the observed trend in our case–control analyses.

One strength of our study is that it was conducted in a homogeneous sample of high-risk women and population controls with detailed lifetime information on mammograms. Our study is unique in that sequencing data for 113 DNA repair genes are available for an important subset of the study population [16]. Indeed, our study is the first to investigate not only the effect of mammograms but also the joint effect of mammogram exposure and D-PDVs in DNA repair genes in women at high risk of BC. We cannot exclude potential biases first due to the classification of the genes according to the ORs calculated in the same population and second due to a classification based on point estimates with, for many of them, insufficient power to characterize precisely their effects. However, the latter potential misclassification should be differential according to gene groups to lead to the observed differential association with mammograms, which is unlikely. Moreover, defining the group of genes with a point estimate falling in a large range of ORs around 1 as the group of genes “not associated with BC” might have an impact on the findings. Therefore, we performed numerous sensitivity analyses performing case-only analysis, changing the boundaries and assuming additivity of the effects of genes, and we found similar trends in the difference in the BC risk between groups.

Finally, the observed difference in the association between mammography and BC risk between the DNA repair gene groups cannot be explained by potential differences in characteristics like age at the diagnostic or educational level or even the number of BC in the family at censure (Appendix A).

Our exploratory findings showed that mammogram exposure (ever vs. never) does not increase BC risk, even among women carrying a rare D-PDV in a DNA repair gene potentially increasing BC risk, except when first exposure occurred before age 30. Surprisingly, a D-PDV in a DNA repair gene potentially decreasing BC risk may increase sensibility to mammograms’ exposure.

The evaluation of low doses in epidemiological studies is challenged by many potential limitations like lack of statistical power, unheeded time dependency, misclassification in exposure assessment, inappropriate evaluation of confounding risk factors, and failure to take hormesis into account [24]. Therefore, we acknowledge that both the lifetime numbers of mammograms as a surrogate for X-ray doses received at breasts and the potential misclassification of the D-PDVs might weaken our observations. However, our study is the first to investigate the joint effect of mammogram exposure and D-PDV in DNA worth checking our findings in further studies on larger populations, ideally with prospectively recorded detailed exposure data and evaluating other genes that could modify radiation sensitivity.

## 5. Conclusions

Our results showed that mammogram exposure (ever vs. never) does not increase BC risk, even among women carrying a D-PDV in a DNA repair gene potentially increasing BC risk, except when first exposure occurred before age 30. Surprisingly, D-PDVs in a DNA repair gene potentially decreasing BC risk may increase sensitivity to mammograms’ exposure. Further studies on larger populations are needed to verify our findings and to evaluate other genes that could modify radiation sensitivity. Even though the current data are clear in demonstrating that mammographic screening thanks to early diagnosis significantly reduces the risk of mortality from BC [25,26], identification of sub-populations that are more or less susceptible to ionizing radiation is much more important and clinically relevant.

## Figures and Tables

**Figure 1 cancers-17-01062-f001:**
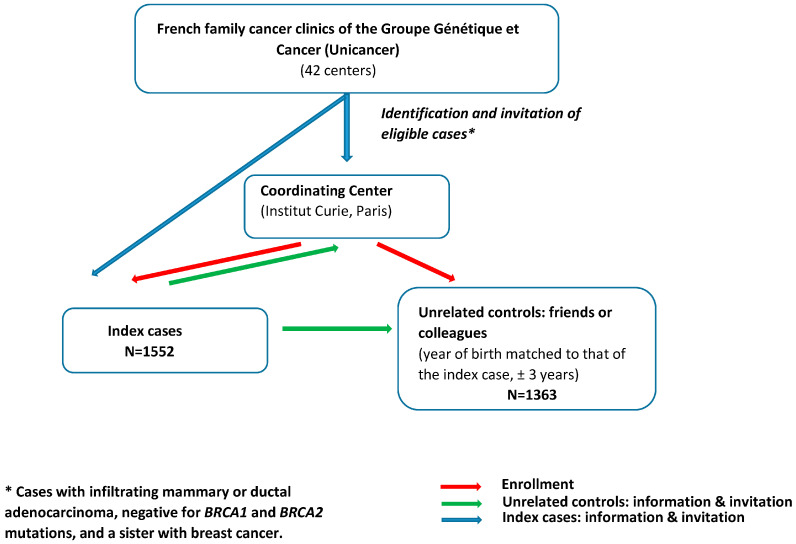
Recruitment process in GENESIS study.

**Figure 2 cancers-17-01062-f002:**
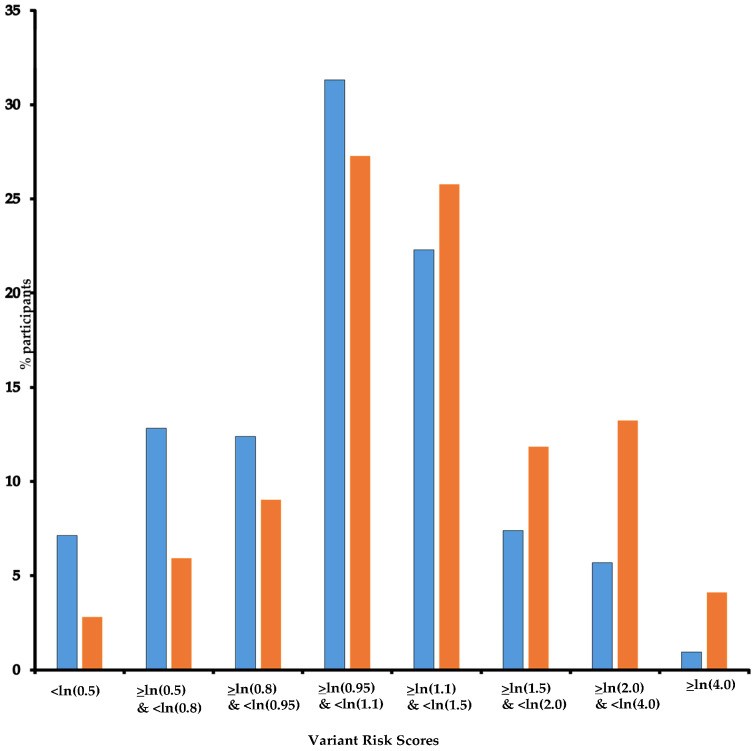
DNA repair Variant Score Risk distribution. Legend: Horizontal axis: Variant Risk Scores; vertical axis: % participants; Blue: controls; Orange: cases. Note: For each participant, a Variant Risk Score (VRS = β_1_x_1_ + β_2_x_2_ +…+ β_113_x_113_) was built with β_i_ per DNA repair gene minor allele mean effect, i.e., ln odds ratio (OR) for breast cancer, and x_i_ equals 1 or 0 for being carrier of the i-th DNA repair gene minor allele or not, respectively.

**Table 1 cancers-17-01062-t001:** Characteristics of GENESIS participants.

Characteristics	Cases N = 1552		Controls N = 1363	
	No.	%	No.	%
**Birth cohort**				
≤1945	488	31.4	294	21.6
1946–1959	797	51.4	706	51.8
≥1960	267	17.2	363	26.6
**Age at censoring, years**				
Mean (SD)	50.2 (9.3)		55.8 (9.9)	
≤45	513	33.1	201	14.8
46–50	336	21.7	197	14.5
51–60	473	30.5	485	35.6
>60	230	14.8	480	35.2
**Education level**				
Intermediate/High	780	50.3	916	67.2
Basic	714	46.0	434	31.8
No graduated	58	3.7	12	0.9
Missing	0	0.0	1	0.1
**Body Mass Index**				
≥18.5 and <25	1019	65.6	869	63.8
<18.5	69	4.5	32	2.3
≥25 and <30	341	22.0	345	25.3
≥30	120	7.7	117	8.6
Missing	3	0.2	0	0.0
**Smoking**				
No	832	53.6	680	49.9
Current	159	10.2	158	11.6
Past	550	35.4	514	37.7
Missing	11	0.7	11	0.8
**Number of full-term pregnancies**				
>2	445	28.7	411	30.2
1–2	921	59.3	763	56.0
0	185	11.9	187	13.7
Missing	1	0.1	2	0.1
**Number of breast cancers in the family** **at censure ^a^**				
None	427	27.5	959	70.4
1	818	52.7	187	13.7
≥2	307	19.8	216	15.9
**Tumor estrogen receptors (ERs)**				
ER+	818	52.7		
ER−	168	10.8		
Missing	566	36.5		
**Total number of variants in DNA repair genes**				
0	191	19.2	241	20.7
1	294	29.5	374	32.2
2	265	26.6	291	25.0
3	151	15.2	160	13.8
4	61	6.1	64	5.5
5	26	2.6	28	2.4
6	5	0.5	4	0.3
7	3	0.3	0	0.0
8	1	0.1	0	0.0
Missing	555	35.8	201	14.8
**DNA repair gene group ^b^**				
Group “Reduced”				
0	723	46.6	721	52.9
≥1	274	17.7	441	32.4
Group “Independent”				
0	575	37.1	659	48.4
≥1	422	27.2	503	36.9
Group “Increased”				
0	422	27.2	667	48.9
≥1	575	37.1	495	36.3

^a^ Excluding one affected sister when all affected sisters were diagnosed with breast cancer before the index case. ^b^ Individuals carrying at least one rare variant in one of the Gene Groups: Group “Reduced” (OR < 0.9); Group “Independent” (0.9 ≤ OR ≤ 1.1); Group “Increased” (OR > 1.1) as defined in Girard et al. [16].

**Table 2 cancers-17-01062-t002:** Mammography exposure characteristics.

Characteristics ^a^	Cases N = 1552		Controls N = 1363	
	No.	%	No.	%
**Mammography**				
Never	266	17.1	102	7.5
Ever	1271	81.9	1252	91.9
Missing	15	1.0	9	0.7
**Lifetime number of mammograms**				
1–4	614	39.6	483	35.4
5–9	409	26.3	537	39.4
≥10	248	16.0	232	17.0
No exposure	266	17.1	102	7.5
Missing	15	1.0	9	0.7
Mean ^b^ (SD ^b^)	6.04 (5.21)		6.25 (4.19)	
**Lifetime number of mammogram views**				
1–9	208	1.4	165	12.1
10–19	198	12.8	227	16.7
20–29	158	10.2	233	17.1
30–39	123	7.9	189	13.9
≥40	206	13.3	230	16.9
No exposure	266	17.1	102	7.5
Missing	392	25.3	217	15.9
Mean ^b^ (SD ^b^)	27.9 (25.2)		27.7 (19.5)	
**Age at first exposure (years)**				
≥50	147	9.5	295	21.6
40–49	487	31.4	618	45.3
30–39	483	31.1	269	19.7
<30	132	8.5	59	4.3
No exposure	266	17.1	102	7.5
Missing	37	2.4	20	1.5
Mean ^b^ (SD ^b^)	38.9 (8.52)		43.1 (8.25)	
**Duration since first exposure (years)**				
≤5	266	17.1	188	13.8
6–10	315	20.3	296	21.7
11–20	448	28.9	506	37.1
>20	220	14.2	251	18.4
No exposure	266	17.1	102	7.5
Missing	37	2.4	20	1.5
Mean ^b^ (SD ^b^)	12.7 (8.28)		13.9 (7.82)	

^a^ Lifetime exposures up to one year prior to diagnosis for cases and up to one year prior to the date of questionnaire completion for controls. ^b^ Among exposed.

**Table 3 cancers-17-01062-t003:** Effect of lifetime mammography exposure (any exposure) on breast cancer risk according to the number of exposures, the age at first exposure, and time since first exposure.

	Number of		OR ^a^	95% CI	Multiple Imputation	
	Cases	Controls			OR ^a^	95% CI
**Mammography exposure**						
Never	266	102	1		1	
Ever	1271	1252	0.99	0.69–1.43	0.98	0.68–1.42
**Number of mammograms**						
0	266	102	1		1	
1–4	614	483	1.04	0.72–1.51	1.02	0.70–1.49
5–9	409	537	0.82	0.54–1.24	0.82	0.55–1.24
≥10	248	232	1.36	0.86–2.16	1.36	0.86–2.16
*Continuous (incl 0)*			*1.04*	*1.01–1.06*		
**Age at first exposure, years** ^b^						
No exposure	266	102	1		1	
≥50	147	295	0.56	0.34–0.91	0.56	0.34–0.91
40–49	457	618	0.67	0.45–0.99	0.67	0.45–0.99
30–39	483	269	1.22	0.82–1.80	1.20	0.81–1.77
<30	132	59	1.52	0.91–2.53	1.46	0.88–2.45

≥50	147	295	1		1	
40–49	457	618	1.20	0.87–1.64	1.20	0.87–1.65
30–39	483	269	2.17	1.49–3.18	2.15	1.47–3.15
<30	132	59	2.71	1.60–4.58	2.63	1.55–4.46
No exposure	266	102	1.79	1.10–2.90	1.79	1.10–2.93
*Continuous (no exposure excluded)*			*0.98*	*0.96–0.99*		
**Time since first exposure, years** ^b^						
No exposure	266	102	0.98	0.65–1.48	0.99	0.66–1.49
≤5	266	188	1		1	
6–10	315	296	0.90	0.64–1.26	0.90	0.64–1.26
11–20	448	506	0.97	0.70–1.35	0.97	0.69–1.34
>20	220	251	1.17	0.77–1.78	1.14	0.75–1.75
*Continuous (no exposure excluded)*			*1.02*	*1.01–1.04*		

Abbreviations: OR (95% CI): odds ratio (95% confidence interval); missing values coded as an additional category. ^a^ Adjusted for age at censoring, birth cohort (≤1945; 1946–1959; ≥1960), number of full-term pregnancies (>2; 1–2; 0), educational level (intermediate/high; basic; no graduated), BMI (18.5–24.9; <18.5; ≥25), smoking (no; current; past), and number of relatives with breast cancer (0; 1; ≥2), chest X-ray exposure (ever vs. never), being eligible for the national screening (yes/no), with missing included in reference categories for each variable when no multiple imputation. Chest X-ray exposure includes pulmonary radiological examinations in the field of preventive/occupational medicine or for lung disease, preoperative radiological examinations, and radiological examinations of heart and thoracic vessels for all the reported procedures. ^b^ Adjusted as ^a^ plus number of mammography (<10; ≥10).

**Table 4 cancers-17-01062-t004:** Effect of lifetime mammography exposure (any exposure) on breast cancer risk according to the number of exposures, the age at first exposure, and time since first exposure by ER status of the breast tumor.

	ER−				ER+			
	Number of				Number of			
	Cases N = 168	Controls N = 1363	OR ^a^	95% CI	Cases N = 818	Controls N = 1363	OR ^a^	95% CI
**Mammography exposure**								
Never	25	102	1		107	102	1	
Ever	140	1252	1.78	0.83–3.83	706	1252	1.42	0.92–2.19
**Number of mammograms**								
0	25	102	1		67	102	1	
1–4	77	483	1.87	0.87–4.00	244	483	1.46	0.94–2.26
5–9	50	537	1.29	0.54–3.07	314	537	1.27	0.77–2.09
≥10	13	232	1.01	0.35–2.92	605	232	2.40	1.38–4.15
*Continuous (incl 0)*			*0.99*	*0.93–1.06*			*1.08*	*1.04–1.11*
**Age at first exposure, years** ^b^								
No exposure	25	102	1		107	102	1	
≥50	17	295	1.21	0.42–3.46	133	295	0.78	0.43–1.41
40–49	57	618	1.35	0.58–3.14	176	618	0.91	0.57–1.47
30–39	55	269	2.19	0.99–4.83	246	269	1.67	1.05–2.65
<30	11	59	1.53	0.55–4.39	138	59	2.33	1.30–4.17
*Continuous (no exposure excluded)*			*0.99*	*0.96–1.03*			*0.97*	*0.95–0.99*
**Time since first exposure, years** ^b^								
No exposure	25	102	0.53	0.23–1.18	107	102	0.71	0.44–1.15
≤5	36	188	1		80	188	1	
6–10	36	296	0.85	0.46–1.57	277	296	0.99	0.67–1.47
11–20	53	506	0.98	0.53–1.79	257	506	1.04	0.71–1.52
>20	15	251	0.74	0.30–1.84	79	251	1.29	0.79–2.09
*Continuous (no exposure excluded)*			*1.00*	*0.97–1.04*			*1.03*	*1.01–1.05*

Abbreviations: OR (95% CI): odds ratio (95% confidence interval); missing values coded as an additional category. ^a^ Adjusted for age at censoring, birth cohort (≤1945; 1946–1959; ≥1960), number of full-term pregnancies (>2; 1–2; 0), educational level (intermediate/high; basic; no graduated), BMI (18.5–24.9; <18.5; ≥25), smoking (no; current; past), number of relatives with breast cancer (0; 1; ≥2), chest X-ray exposure (ever vs. never), being eligible for the national screening (yes/no), with missing included in reference categories for each variable. Chest X-ray exposure includes pulmonary radiological examinations in the field of preventive/occupational medicine or for lung disease, preoperative radiological examinations, and radiological examinations of heart and thoracic vessels for all the reported procedures. ^b^ Adjusted as ^a^ plus number of mammography (<10; ≥10).

**Table 5 cancers-17-01062-t005:** Main effect of variant carrier status on breast cancer.

DNA Repair Rare Variants					Multiple Imputation for the Adjustment Variables	
	Number of					
	Cases	Controls	OR ^a^	95% CI	OR ^a^	95% CI
**“Reduced” ^b^**						
No	723	721	1		1	
Yes	274	441	0.62	0.50–0.77	0.62	0.50–0.77
Number of variants						
0	723	721	1		1	
1	227	341	0.66	0.53–0.84	0.67	0.53–0.84
2	44	85	0.52	0.34–0.81	0.51	0.33–0.79
≥3	3	15	0.26	0.07–1.03	0.25	0.06–1.01
*Continuous*			*0.69*	*0.59–0.81*	*0.69*	*0.59–0.81*
**“Independent” ^c^**						
No	575	659	1		1	
Yes	422	503	1.02	0.84–1.25	1.03	0.84–1.25
Number of variants						
0	575	659	1		1	
1	309	389	0.97	0.78–1.21	0.97	0.78–1.21
2	96	92	1.16	0.81–1.67	1.17	0.82–1.67
≥3	17	22	1.32	0.62–2.80	1.32	0.62–2.77
*Continuous*			*1.07*	*0.93–1.22*	*1.06*	*0.93–1.22*
**“Increased” ^d^**						
No	422	667	1		1	
Yes	575	495	1.98	1.62–2.42	1.98	1.62–2.42
Number of variants						
0	422	667	1		1	
1	381	357	1.86	1.49–2.32	1.81	1.45–2.26
2	142	116	1.99	1.44–2.74	1.95	1.42–2.68
≥3	52	22	3.91	2.15–7.08	3.88	2.15–7.01
*Continuous*			*1.52*	*1.34–1.71*	*1.51*	*1.34–1.71*

Abbreviations: OR (95% CI): odds ratio (95% confidence interval). ^a^ Adjusted for age at censoring, birth cohort (≤1945; 1946–1959; ≥1960), number of full-term pregnancies (>2; 1–2; 0), mammography use (never; ever), educational level (intermediate/high; basic; not graduated), BMI (18.5–24.99; <18.5; ≥25 and <30; ≥30), smoking (no; current; past), and other DNA repair genes groups. ^b^ At least one variant in a gene from the “Reduced” group. ^c^ At least one variant in a gene from the “Independent” group. ^d^ At least one variant in a gene from the “Increased” group.

**Table 6 cancers-17-01062-t006:** Effect of lifetime mammography exposure (any exposure) on breast cancer risk according to the number of exposures, the age at first exposure, and time since first exposure by DNA repair gene group.

	“Reduced”				“Independent”				“Increased”					
	Number of				Number of				Number of				P_int_	
	Cases N = 274	Controls N = 441	OR ^a^	95% CI	Cases N = 422	Controls N = 503	OR ^a^	95% CI	Cases N = 575	Controls N = 495	OR ^a^	95% CI	“Independent”	“Increased”
**Mammography exposure**														
Never	19	32	1		44	29	1		62	28	1			
Ever	251	406	2.17	0.92–5.15	377	471	1.25	0.62–2.54	509	464	0.92	0.48–1.75	ns	0.021
**Number of** **mammograms**														
0	19	32	1		44	29	1		62	28	1			
1–4	105	149	2.27	0.95–5.41	158	162	1.34	0.66–2.74	207	158	0.99	0.52–1.90	ns	0.020
5–9	92	178	1.77	0.68–4.65	131	219	1.04	0.47–2.30	180	212	0.74	0.36–1.52	ns	0.019
≥10	54	79	2.34	0.81–6.74	88	90	2.30	0.96–5.54	122	94	1.49	0.67–3.30	ns	ns
*Continuous incl 0*			*1.03*	*0.98–1.08*			*1.06*	*1.02–1.10*			*1.06*	*1.02–1.10*		
**Age at first exposure, years** ^b^														
No exposure	19	32	1		44	29	1		62	28	1			
≥50	29	100	1.65	0.53–5.08	40	111	1.01	0.39–2.62	62	111	0.75	0.32–1.73	ns	0.019
40–49	104	195	1.77	0.70–4.46	154	239	0.93	0.43–2.02	193	223	0.76	0.38–1.53	ns	0.021
30–39	90	88	2.22	0.91–5.54	138	97	1.48	0.70–3.11	196	102	0.99	0.50–1.94	ns	0.057
<30	25	18	3.16	1.01–9.89	42	23	1.61	0.64–4.01	52	21	1.20	0.51–2.80	ns	ns
*Continuous (no exposure excluded)*			*1.00*	*0.97–1.05*			*0.99*	*0.96–1.02*			*0.99*	*0.96–1.01*		
**Time since first** **exposure, years** ^b^														
No exposure	19	32	0.39	0.16–0.99	44	29	0.69	0.32–1.51	62	28	0.98	0.48–2.01	0.017	0.008
≤5	51	61	1		62	65	1		79	58	1			
6–10	60	91	0.74	0.37–1.47	90	111	0.83	0.45–1.51	132	108	0.84	0.48–1.46	ns	ns
11–20	98	166	0.78	0.40–1.53	148	191	0.87	0.49–1.55	190	188	0.94	0.54–1.62	ns	ns
>20	39	83	0.55	0.23–1.33	74	100	0.83	0.40–1.72	102	103	0.99	0.51–1.94	ns	0.051
*Continuous (no exposure excluded)*			*0.99*	*0.96–1.03*			*1.01*	*0.98–1.04*			*1.01*	*0.99–1.04*		

Abbreviations: OR (95% CI): odds ratio (95% confidence interval); missing values coded as an additional category. ^a^ Adjusted for age at censoring, birth cohort (≤1945; 1946–1959; ≥1960), number of full-term pregnancies (>2; 1–2; 0), educational level (intermediate/high; basic; no graduated), BMI (18.5–24.9; <18.5; ≥25), smoking (no; current; past), chest X-ray exposure (ever vs. never), being eligible for the national screening (yes/no), number of relatives with breast cancer (0; 1; ≥2), and the two other DNA repair genes groups, with missing included in reference categories for each variable. Chest X-ray exposure includes pulmonary radiological examinations in the field of preventive/occupational medicine or for lung disease, preoperative radiological examinations, and radiological examinations of heart and thoracic vessels for all the reported procedures. ^b^ Adjusted as ^a^ plus number of mammography (<10; ≥10).

**Table 7 cancers-17-01062-t007:** Effect of mammography exposure according to the DNA repair Variant Score Risk.

	Number of		OR ^a^	95% CI	*p*
	Cases	Controls			
**Mammography exposure**					
No exposure	92	74	1		
Ever and VRS < ln(0.8)	83	214	2.56	0.94–6.94	
Ever and ln(0.8) ≤ VRS < ln(0.95)	82	134	3.88	1.39–10.8	
Ever and ln(0.95) ≤ VRS < ln(1.1)	247	332	2.29	1.07–4.92	P_het_ = 0.104
Ever and ln(1.1) ≤ VRS < ln(1.5)	232	243	1.24	0.54–2.87	P_int_ ^b^ = 0.006
Ever and VRS ≥ ln(1.5)	254	157	0.30	0.10–0.87	
**Age at first exposure, years**					
No exposure	92	74	1		
≥40 and VRS < ln(0.95)	87	256	1.41	0.57–3.48	
≥40 and ln(0.95) ≤ VRS < ln(1.1)	124	245	1.05	0.50–2.19	
≥40 and ln(1.1) ≤ VRS < ln(1.5)	122	180	0.74	0.33–1.66	ns
≥40 and VRS ≥ ln(1.5)	126	109	0.33	0.14–0.82	
<40 and VRS < ln(0.95)	76	90	2.60	1.03–6.55	
<40 and ln(0.95) ≤ VRS < ln(1.1)	116	83	2.36	1.10–5.05	
<40 and ln(1.1) ≤ VRS < ln(1.5)	108	62	1.59	0.68–3.71	ns
<40 and VRS ≥ ln(1.5)	126	44	0.64	0.25–1.60	
**Variant Risk Score**					
<ln(0.8)	28	83	1		
≥ln(0.8) and <ln(0.95)	149	293	1.71	0.96–3.02	
≥ln(0.95) and <ln(1.1)	272	364	2.29	1.32–3.99	
≥ln(1.1) and <ln(1.5)	257	259	3.12	1.78–5.46	
≥ln(1.5)	291	163	5.71	3.23–10.1	

Abbreviations: *p*: *p* value; P_het_: P heterogeneity; P_int_: P interaction; ^a^ Adjusted for age at censoring, birth cohort (≤1945; 1946–1959; ≥1960), number of full-term pregnancies (>2; 1–2; 0), educational level (intermediate/high; basic; no graduated), BMI (18.5–24.9; <18.5; ≥25), smoking (no; current; past), chest X-ray exposure (ever vs. never), being eligible for the national screening (yes/no), number of relatives with breast cancer (0; 1; ≥2), and VRS. Chest X-ray exposure includes pulmonary radiological examinations in the field of preventive/occupational medicine or for lung disease, preoperative radiological examinations, and radiological examinations of heart and thoracic vessels for all the reported procedures. ^b^ Interaction was tested with VRS as a continuous variable.

**Table 8 cancers-17-01062-t008:** Effect of lifetime mammography exposure (any exposure) on breast cancer risk according to the number of exposures, the age at first exposure, and time since first exposure by DNA repair gene group (Multiple Imputation).

	“Reduced”				“Independent”				“Increased”			
	Number of				Number of				Number of			
	Cases N = 274	Controls N = 441	OR ^a^	95% CI	Cases N = 422	Controls N = 503	OR ^a^	95% CI	Cases N = 575	Controls N = 495	OR ^a^	95% CI
**Mammography** **exposure**												
Never	19	32	1		44	29	1		62	28	1	
Ever	251	406	2.11	0.89–5.02	377	471	1.22	0.61–2.50	509	464	0.89	0.46–1.70
**Number of** **mammograms**												
0	19	32	1		44	29	1		62	28	1	
1–4	52	81	2.25	0.89–5.66	63	84	1.10	0.50–2.39	87	83	0.93	0.45–1.62
5–9	51	97	1.83	0.71–4.69	82	112	1.21	0.56–2.60	103	111	0.73	0.37–1.45
≥10	98	205	2.46	0.93–6.51	155	243	1.56	0.72–3.40	220	236	1.37	0.67–2.82
**Age at first exposure, years** ^b^												
No exposure	19	32	1		44	29	1		62	28	1	
≥50	29	100	1.52	0.60–4.89	40	111	0.75	0.30–1.85	62	111	0.64	0.28–1.47
40–49	104	195	1.72	0.68–4.35	154	239	0.81	0.38–1.74	193	223	0.66	0.33–1.32
30–39	90	88	2.12	0.85–5.29	138	97	1.36	0.65–2.87	196	102	0.88	0.44–1.74
<30	25	18	2.85	0.90–9.04	42	23	1.46	0.58–3.68	52	21	0.97	0.41–2.30
**Time since first** **exposure, years** ^b^												
No exposure	51	61	0.42	0.16–1.05	44	29	0.76	0.35–1.65	62	28	1.06	0.51–2.18
≤5	19	32	1		62	65	1		79	58	1	
6–10	60	91	0.70	0.35–1.43	90	111	0.78	0.42–1.45	132	108	0.75	0.42–1.34
11–20	98	166	0.70	0.33–1.50	148	191	0.82	0.43–1.58	190	188	0.76	0.41–1.40
>20	39	83	0.56	0.23–1.39	74	100	1.08	0.35–1.65	102	103	1.00	0.50–1.98

Abbreviations: OR (95% CI): odds ratio (95% confidence interval); missing values coded as an additional category. ^a^ Adjusted for age at censoring, birth cohort (≤1945; 1946–1959; ≥1960), number of full-term pregnancies (>2; 1–2; 0), educational level (intermediate/high; basic; no graduated), BMI (18.5–24.9; <18.5; ≥25), smoking (no; current; past), chest X-ray exposure (ever vs. never), being eligible for the national screening (yes/no), number of relatives with breast cancer (0; 1; ≥2), and the two other DNA repair genes groups. Chest X-ray exposure includes pulmonary radiological examinations in the field of preventive/occupational medicine or for lung disease, preoperative radiological examinations, and radiological examinations of heart and thoracic vessels for all the reported procedures. ^b^ Adjusted as ^a^ plus number of mammography (<10; ≥10).

## Data Availability

The data underlying this article will be shared upon reasonable request to the corresponding author.

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
