# Peer review of "Screening Mammography and Breast Cancer: Variation in Risk with Rare Deleterious or Predicted Deleterious Variants in DNA Repair Genes"

_cancers, 2025, doi:10.3390/cancers17071062_

Round 1

Reviewer 1 Report

Comments and Suggestions for Authors

The manuscript submitted by Nadine Andrieu and team investigate the joint effect of mammogram exposure and variant in DNA repair genes excluding BRCA1 and BRCA2 in women at high risk of breast cancer.   Identification of populations that are susceptible to ionizing radiation may be clinically relevant even though its clinically proven that mammographic screening reduces the risk of mortality from breast cancer. Overall, this is an interesting study and may have clinical relevance to the population having mutations in DNA repair genes.  However, the authors should make these suggested changes to manuscript to provide readers with sufficient information to understand the results.

  • Authors should check the cover page for any grammatical mistake very carefully. Page 16, Nadine Andrieu and PhD should be Nadine Andrieu, PhD

Introduction:

  • Authors needs to add more about how the panel of the DNA repair genes carrying a rare deleterious or predicted deleterious variants (D-PDV) (other than BRCA1/2) were selected to test association between mammography screening exposures to breast cancer risk. Selected DNA repair gene does not provide sufficient information to the readers to understand the results. This background knowledge to be placed in introduction will make it easy to understand.

Results:

  • Table 1: It is hard to understand the Table as in column Cases No and % are not aligned properly. Authors should align the No. and % so that it is easy to understand. Check all other Tables for the correct alignment as there are lot of typographical and alignment errors in the manuscript.
  • Figure 2: DNA repair Variant Score Risk distribution.\

Revise the figure so that it is clear, Y axis is not labeled. Colored bar will be clearer to understand the figure.  Figure 2. DNA repair Variant Score Risk distribution.\, Is there a reason for \.

Additionally, Line 480, check for typographical mistake. It should be Availability of data and materials. There are many typographical and corrections which the authors need to review and revise the manuscript very carefully to publish in Cancers.

Author Response

Reponse to reviewer 1’s comments and suggestions

First of all, we thank the reviewer for the encouraging comments. We took all his/her comments /suggestions into account.

Comment 1:

  • Authors should check the cover page for any grammatical mistake very carefully. Page 16, Nadine Andrieu and PhD should be Nadine Andrieu, PhD

Response: We checked the cover page and corrected grammatical mistake.

Comment 2:

Introduction:

  • Authors needs to add more about how the panel of the DNA repair genes carrying a rare deleterious or predicted deleterious variants (D-PDV) (other than BRCA1/2) were selected to test association between mammography screening exposures to breast cancer risk. Selected DNA repair gene does not provide sufficient information to the readers to understand the results. This background knowledge to be placed in introduction will make it easy to understand.

Response: We acknowledge that « selected DNA repair gene » in the introduction is very vague and we added some information:

« Furthermore, we evaluated whether carrying a rare deleterious or predicted deleterious variants (D-PDV) (i.e. loss-of-function or missense variant with a predicted deleterious effect, phred CADD score > 20) in 113 DNA repair genes (other than BRCA1/2) previously selected, modifies the association between mammography screening exposures and BC risk. »

Comment 3:

  • Table 1: It is hard to understand the Table as in column Cases No and % are not aligned properly. Authors should align the No. and % so that it is easy to understand. Check all other Tables for the correct alignment as there are lot of typographical and alignment errors in the manuscript.

Response: We apologize for the misalignment of numbers in some tables and we did our best to better align the numbers in the tables.

Comment 4:

  • Figure 2: DNA repair Variant Score Risk distribution.\

Revise the figure so that it is clear, Y axis is not labeled. Colored bar will be clearer to understand the figure.  Figure 2. DNA repair Variant Score Risk distribution.\, Is there a reason for \.

Response: We added Y axis label and colored bars in figure 2. «  \ »  had no reason, a typo and we deleted it. 

Comment 5:

Additionally, Line 480, check for typographical mistake. It should be Availability of data and materials. There are many typographical and corrections which the authors need to review and revise the manuscript very carefully to publish in Cancers.

Response: « Availability of data and materials » has been changed by « Data Availability Statement » as suggested by « Cancers ». Additionally we reviewed and corrected any typographical mistakes.

Reviewer 2 Report

Comments and Suggestions for Authors

The authors present an interesting study on the potential association between screening mammograms and the presence of deletoriuos variants of DNA repair genes with respect to increasing breast cancer risk. The study is well written, easy to follow, comprising detailed information on the methodology and results obtained from the analysis. The authors provide explanations on the potential limitations and biases that could arise from the study design. However, very limited data is present in the literature with regard to the risk of developing breast cancer as a result of repeated mammograms, especially in potentially high risk populations. In my opinion, the data presented in the current manuscript offers new data that can be used for the development of future screening indications and further studies for a more in-depth analysis of mammographic effects in certain high-risk groups. 

Some minor comments:

-in the methods section it would useful to mention the study interval (years)

— in table 1, regarding the number of full-term births, the first category is equal or greater than 2, whereas the second category is 1-2 births. Cpuld you clarify this, as the overlap between these to groups can be confusing for readers. 

Author Response

Reponse to reviewer 2’s comments and suggestions

First of all, we thank the reviewer for the encouraging comments. We took all his/her comments into account.

Comment 1

-in the methods section it would useful to mention the study interval (years)

Response: Inclusion of GENESIS participants started in February 2007 and ended in December 2013. We have added this information in the methods section.

Comment 2

— in table 1, regarding the number of full-term births, the first category is equal or greater than 2, whereas the second category is 1-2 births. Could you clarify this, as the overlap between these to groups can be confusing for readers. 

Response: The first category > 2 is a typo mistake, in fact this category is >2 and has been corrected in the table.

Reviewer 3 Report

Comments and Suggestions for Authors

This manuscript has an insightful analysis of the impact of mammographic screening in individuals carrying rare deleterious DNA repair gene variants. The study design is robust, and the statistical approach ensures the reliability of the findings. The discussion presented a comprehensive implications of these results for breast cancer screening protocols.

Please:

  1. Explain whether some genes in the "reduced" risk group work in a special way that affects how they interact with mammography exposure.
  2. Emphasize any clinical recommendations based on the findings.

Author Response

Reponse to reviewer 3’s comments and suggestions

First of all, we thank the reviewer for the encouraging comments. We took all his/her comments into account.

This manuscript has an insightful analysis of the impact of mammographic screening in individuals carrying rare deleterious DNA repair gene variants. The study design is robust, and the statistical approach ensures the reliability of the findings. The discussion presented a comprehensive implications of these results for breast cancer screening protocols.

Comment 1

Explain whether some genes in the "reduced" risk group work in a special way that affects how they interact with mammography exposure.

Response: We do not yet know the functional impact of the “reduced” risk D-PDVs and their potential consequence on the DNA repair function after being exposed to mammography. A possible hypothetical explanation is that the mammary cells of women carrying a D-PDV in DNA repair gene associated with an “increased” risk of BC would be less able to repair damages due to mammography ionizing radiations and therefore would have a greater propensity to go into apoptosis than the mammary cells of women carrying a DNA repair variant associated with a reduced risk, i.e. more suitable to repair DNA damaging and therefore to accumulate oncogenic repair errors. It would be very interesting to carry out functional studies of these D-PDVs before and after ionizing irradiation to observe the consequences on the cells’ ability to repair DNA damaging. 

Comment 2

Emphasize any clinical recommendations based on the findings.

Response: We thank the reviewer for this suggestion and emphasize potential clinical relevance in the discussion paragraph. However, our results are very preliminary and need to be verified and further investigated before proposing recommendations on possible adaptations of screening or diagnostic radiological examinations based on genetic susceptibilities. However we added a conclusion section where we emphasize on potential clinical relevance of our findings.

« Our results showed that mammogram exposure (ever versus never) does not increase BC risk, even among women carrying a D-PDV in a DNA repair gene potentially increasing BC risk, except when first exposure occurred before age 30. Surprisingly, D-PDV in a DNA repair gene potentially decreasing BC risk may increase sensitivity to mammograms exposure…..further studies on larger populations are needed to verify our findings and to evaluate other genes that could modify radiation sensitivity. Even though the current data are clear in demonstrating that mammographic screening thanks to early diagnosis significantly reduces the risk of mortality from BC [22,23], identification of sub-populations that are more or less susceptible to ionizing radiation is much important and possibly, clinically relevant. »

Reviewer 4 Report

Comments and Suggestions for Authors

In this paper, the authors investigated the risk of breast cancer associated with additional exposure to mammograms. This is a very interesting finding that could help improve patient care in clinical settings. However, I believe the paper has several areas that need revision to enhance readability. Below are my specific comments:

1. Lines 75–77: The combination of these two sentences is confusing. I have a general idea of what the authors are trying to convey, but they need to rephrase this section for better clarity and readability.

2. Lines 193–194: The authors should include the R² value and p-value to properly evaluate the curve-fitting results.

3. Lines 202–203: The type of statistical test used needs to be specified—was it a t-test, Z-test, or another test?

4. Lines 225–227: The calculation of the reported 4% is unclear. The authors should clearly indicate what the dependent and independent variables are in the equation to help guide the reader.

5. Tables: Please align the numbers for better readability. Additionally, the authors use both “controls” and “ctrls” inconsistently throughout the manuscript. It would be best to maintain consistency in terminology.

6. Table 3: The distinction between “Number of mammograms” and “Time since first exposure” is unclear. The authors should clarify how these variables differ and ensure the terminology is precise.

Comments on the Quality of English Language

The language need to be improved to increase readability. 

Author Response

Reponse to reviewer 4’s comments and suggestions

First of all, we thank the reviewer for the comments. We took all his/her comments /suggestions into account.

  1. Lines 75–77: The combination of these two sentences is confusing. I have a general idea of what the authors are trying to convey, but they need to rephrase this section for better clarity and readability.

Response: We rephrased this section: « Participants reported their lifetime mammography exposures in a questionnaire. Additionally, germline rare deleterious or predicted deleterious variants (D-PDV) were investigated in 113 DNA repair genes. No association was found between having been exposed to mammograms (never vs. ever) and BC risk. However, when considering the number of mammograms, an increase in BC risk of 4% (95%CI:1%-6%) per additional exposure was found. »

  1. Lines 193–194: The authors should include the R² value and p-value to properly evaluate the curve-fitting results.

Response:

We acknowledge that R² is an appropriate measure of fit for linear regression models, but not so much for logistic regression models. We could use a pseudo-R² providing an approximation to assess the fit of logistic regression models; however a high value does not guarantee good predictive ability and should be complemented by other metrics. We believe that adding this metric would not improve the understanding of our results, which are from an exploratory study whose preliminary results need to be checked by further studies.

  1. Lines 202–203: The type of statistical test used needs to be specified—was it a t-test, Z-test, or another test?

Response: we used Z-test and added this information.

  1. Lines 225–227: The calculation of the reported 4% is unclear. The authors should clearly indicate what the dependent and independent variables are in the equation to help guide the reader.

Response: We tested the effect of the number of mammograms on the BC risk using a logistic regression with BC status (case or control) as dependent variable and the number of mammograms (continuous variable) as independent variable (cf. table 3). Then OR=1.04, i.e. 4% increase per additional mammogram. We rephrased this section.

  1. Tables: Please align the numbers for better readability. Additionally, the authors use both “controls” and “ctrls” inconsistently throughout the manuscript. It would be best to maintain consistency in terminology.

Response: We apologize for the misalignment of numbers in some tables and we have done our best to better align the numbers in the tables. We have also consistently used « controls » throughout the manuscript.

  1. Table 3: The distinction between “Number of mammograms” and “Time since first exposure” is unclear. The authors should clarify how these variables differ and ensure the terminology is precise.

Response: Variables considered in the analyses were: ever vs. never exposed, lifetime number of screening mammograms, age at first mammography, and timing since the first mammograph, i.e. the delay in years between the age at first mammography and the age at censure. This has been added in the method section.

Round 2

Reviewer 4 Report

Comments and Suggestions for Authors

The authors have addressed all my comments, and I think it is good to go